# Targeting Viral and Cellular Cysteine Proteases for Treatment of New Variants of SARS-CoV-2

**DOI:** 10.3390/v16030338

**Published:** 2024-02-22

**Authors:** Davide Gentile, Lucia Chiummiento, Alessandro Santarsiere, Maria Funicello, Paolo Lupattelli, Antonio Rescifina, Assunta Venuti, Anna Piperno, Maria Teresa Sciortino, Rosamaria Pennisi

**Affiliations:** 1Department of Chemistry, Materials and Chemical Engineering “G. Natta”, Politecnico di Milano, Via Mancinelli 7, 20131 Milano, Italy; 2Department of Scienze, University of Basilicata, Viale dell’Ateneo Lucano 10, 85100 Potenza, Italy; alessandro.santarsiere@unibas.it (A.S.); maria.funicello@unibas.it (M.F.); 3Department of Chimica, Sapienza University of Roma, p. le Aldo Moro 5, 00185 Roma, Italy; paolo.lupattelli@uniroma1.it; 4Department of Drug and Health Sciences, University of Catania, V. le A. Doria, 95125 Catania, Italy; arescifina@unict.it; 5International Agency for Research on Cancer (IARC), World Health Organization, 69366 LYON CEDEX 07, France; assuntavenuti@gmail.com; 6Department of Chemical, Biological, Pharmaceutical and Environmental Science, University of Messina, Viale Ferdinando Stagno d’Alcontres 31, 98166 Messina, Italy; apiperno@unime.it (A.P.); mtsciortino@unime.it (M.T.S.)

**Keywords:** SARS-CoV-2, cysteine protease, pseudovirus technology

## Abstract

The continuous emergence of SARS-CoV-2 variants caused the persistence of the COVID-19 epidemic and challenged the effectiveness of the existing vaccines. The viral proteases are the most attractive targets for developing antiviral drugs. In this scenario, our study explores the use of HIV-1 protease inhibitors against SARS-CoV-2. An in silico screening of a library of HIV-1 proteases identified four anti-HIV compounds able to interact with the 3CL^pro^ of SARS-CoV-2. Thus, in vitro studies were designed to evaluate their potential antiviral effectiveness against SARS-CoV-2. We employed pseudovirus technology to simulate, in a highly safe manner, the adsorption of the alpha (α-SARS-CoV-2) and omicron (ο-SARS-CoV-2) variants of SARS-CoV-2 and study the inhibitory mechanism of the selected compounds for cell–virus interaction. The results reported a mild activity against the viral proteases 3CL^pro^ and PL^pro^, but efficient inhibitory effects on the internalization of both variants mediated by cathepsin B/L. Our findings provide insights into the feasibility of using drugs exhibiting antiviral effects for other viruses against the viral and host SARS-CoV-2 proteases required for entry.

## 1. Introduction

The constant evolution of SARS-CoV-2 due to spike (S) protein mutation is of great concern [1]. Four years after the start of the first coronavirus pandemic, several SARS-CoV-2 variants were diversified to transmissibility, severity, and immune evasion [2]. From the emergence of COVID-19 in 2019, periods of apparent evolutionary stasis followed an increase in divergence within significant SARS-CoV-2 lineages. Currently, the co-circulation of the descendants of the omicron BA.2 variant and the variant subjected to monitoring, such as CH.1.1, continues [3]. The distinct phenotypic characteristics, particularly the transmissibility, play a role in the SARS-CoV-2 evolution. In this outbreak scenario, to contain the infection, the scientific community suggested strong social restraint measures and the active development of vaccines. The rapid mutation rate of the virus and the actual findings, which suggested that XBB, the omicron sub-variants of SARS-CoV-2 spreading from 2022, emerged through the recombination of two cocirculating BA.2 lineages, demonstrated the urgent need to develop effective therapeutic strategies. The emergence of new variants and the continued tracking of their evolution do not temporally correlate with the production of new vaccines or their clinical trial evaluation. Therefore, together with the development of new vaccines and new antiviral drugs, the modification of existing vaccines and the repurposing of drugs with a known mechanism of action, toxicology information, and pharmacodynamics profiles can help to develop a sustainable response that impacts the evolution of the virus. The in vitro inhibitory activity of selected compounds against the recombinant protease of HIV has been previously reported by our group [4,5]. Besides, the efficacy of HIV protease inhibitors (PIs), lopinavir and ritonavir, has been investigated against SARS-CoV-2 [6], considering their documented activity against related coronaviruses [7,8]. The study focused on the main protease M^pro^ of SARS-CoV-2 as a potential target for these PIs. While in silico screening identified nelfinavir as a potential M^pro^ inhibitor [9], lopinavir and ritonavir were suggested by molecular dynamics simulation [10]. Previous studies showed the effectiveness of lopinavir/ritonavir against SARS-associated coronavirus [7,8], but recent clinical trials for COVID-19 yielded no significant benefits [11,12,13]. Mahdi and collaborators tested a panel of HIV PIs against SARS-CoV-2 M^pro^ using a cell culture-based model and determined IC_50_ values. The combination of lopinavir plus ritonavir showed the lowest IC_50_, although with cellular viability concerns. Darunavir and atazanavir required higher concentrations but exhibited no cytotoxicity. The study suggested the limited clinical potential for HIV PIs in treating COVID-19, raising the possibility of other molecular targets for these drugs. Thus, we studied their role against proteases employed by SARS-CoV-2 during the infection. The replication machinery of coronaviruses includes the papain-like protease (PL^pro^) and the 3-chymotrypsin-like protease (3CL^pro^), also known as the main protease Mpro, making them highly promising targets for drug design [4,5]. Moreover, considering the results of recent scientific literature about the ability of some viral inhibitors to target both viral and cellular proteases hindering two different steps of the replicative cycle, we also evaluated our drugs as virus entry inhibitors [14,15].

The entry of SARS-CoV-2 into target cells requires the activation of S by host proteases [16]. The host transmembrane serine protease 2, TMPRSS2, and cysteine proteases cathepsin B/L (CatB/CatL) activate S, allowing SARS-CoV-2 entry [17,18,19]. Blocking the cellular proteases prevents SARS-CoV-2 entry and can represent an antiviral strategy alternative to targeting viral proteases. Thus, to simulate the SARS-CoV-2 infection and facilitate the study of high-risk and highly pathogenic enveloped viruses in a biosafety level 2 (BSL-2) environment, we employed a luciferase-expressing pseudovirus encoding the SARS-CoV-2 S protein. The pseudovirus system plays a significant role in studying the mechanism of virus binding and recognition with cell receptors. It allows the screening of potential antiviral drugs targeting the entry phase of the viral cycle [20,21,22]. Selected compounds were assessed by using entry assays to investigate their potential inhibition of the adsorption of the α-SARS-CoV-2 and ο-SARS-CoV-2 pseudoviruses, with a focus on the serine protease TMPRSS2 and cysteine proteases CatB. For this purpose, our strategy included: (i) a computational approach to select potential drugs against SARS-CoV-2 from a library of HIV-1 protease inhibitors, (ii) the identification of 3CL^pro^ inhibitors by enzymatic assays, and (iii) screening for the entry mechanism of a-SARS-CoV-2 and o-SARS-CoV-2 variants in response to treatment with selected compounds (Figure 1). By using a pharmacophore model established in a prior study conducted by our team [23], constructed through the Pharmit server (http://pharmitcsb.pitt.edu/ accessed on 2 April 2023) [24] based on the SARS-CoV-2 3CL^pro^ (PDB ID: 6LU7) and incorporating the complexed structure of the N3 ligand (PRD_002214) as an input, a virtual screening was executed on a collection of HIV-1 inhibitors employing AutoDock4 software [23,24]. The four molecules showing the best binding energy (cutoff −6.5 kcal/mol) were subjected to a 10 ns Molecular Dynamics (MD) simulation study to confirm the stability of the complex. The selection of computational results was guided by considering the binding energies of the investigated compounds with the main viral targets. The two viral proteases (3CL^pro^ and PL^pro^) have been the subject of in silico studies and in vitro biological tests. These studies suggested that our compounds can act as inhibitors of cysteine cathepsin proteases; consequently, an exhaustive work of molecular docking within the active site of CatB and CatL was carried out to elucidate the interactions of our compound with these cellular membrane proteases.

## 2. Materials and Methods

### 2.1. Cells

VERO cell lines (American Type Culture Collection) were cultured in minimal essential medium (EMEM) supplemented with 6% fetal bovine serum (FBS) from Lonza, Belgium. HEK-293T cells were maintained in Dulbecco’s Modified Eagle’s Medium (DMEM Lonza, Verviers, Belgium) supplemented with 10% FBS, 100 U/mL penicillin, and 100 mg/mL streptomycin. All cell lines were grown at 37 °C in a 5% CO_2_ incubator.

### 2.2. Materials

The compounds employed in this study (Figure 1) were synthesized as previously reported [25,26]: *N*-[(2*S*,3*R*)-4-{[(3,4-dimethoxyphenyl)sulfonyl](2-methylpropyl)amino}-3-hydroxy-1-phenylbutan-2-yl]-1*H*-indole-5-carboxamide (IBuDM), *N*-[(2*S*,3*R*)-4-{benzyl[(3,4-dimethoxyphenyl)sulfonyl]amino}-3-hydroxy-1-phenylbutan-2-yl]-1*H*-indole-5-carboxamide (IBnDM), *N*-[(2*S*,3*R*)-4-{benzyl[(3,4-dimethoxyphenyl)sulfonyl]amino}-3-hydroxy-1-phenylbutan-2-yl]-1-benzothiophene-5-carboxamide (BTBnDM), and *N*-[(2*S*,3*R*)-4-{benzyl[(3,4-dimethoxyphenyl)sulfonyl]amino}-3-hydroxy-1-phenylbutan-2-yl]-1-benzofuran-5-carboxamide (BFBnDM). All compounds were solubilized in DMSO.

### 2.3. SARS-CoV-2 3CL^pro^ Pro and PL^pro^ Luminescent Assays

The inhibitory activities of the IBuDM, IBnDM, BTBnDM, and BFBnDM against 3CL^pro^ and PL^pro^ were determined using the SARS-CoV-2 3CL^pro^ and PL^pro^ Luminescent Assays as reported by manufacturer instructions (CS331201-SARS-CoV-2 3CLpro and PLpro Luminescent Assays, Promega, Madison, WI, USA). Purified recombinant SARS CoV-2 3CL^pro^ (9000 µg/mL) (BPS Bioscience, San Diego, CA, USA, cat# 100823) and purified GST-PL^pro^ (1000 nM) (R&D Systems, Minneapolis, MN, USA, Cat# E-611-050) were used. The compounds were tested at 500 µM, GC376 was used as a positive control at 2 µM and DMSO was included as a solvent control at 0.5%. The assay was performed in opaque white 96-well plates with a final volume of 50 μL. Briefly, 12.5 µL of 3CL^pro^ (a final concentration of 16 µg/mL) was incubated for 60 min at 37 °C with 12.5 µL of assay buffer (50 mM HEPES pH 7.2, 10 mM DTT, and 0.1 mM EDTA) containing compounds at final concentration of 500 µM. In the untreated sample, 12.5 µL of 3CL^pro^ was incubated with 12.5 µL of assay buffer. An amount of 25 µL of substrate was then added at final concentration of 40 μM. Reactions were then blocked by adding 50 µL of Luciferin Detection Reagent (Promega cat# V8920), and after 20 min at room temperature, luminescence was read on a GloMax^®^ luminometer. Similarly, 12.5 µL of PL^pro^ (a final concentration of 10 nM) was incubated for 30 min at 25 °C with 12.5 µL of compounds in the assay buffer. The substrate was then added at a final concentration of 40 μM. Reactions were blocked by adding 50 µL of Luciferin Detection Reagent (Promega cat# V8920), and after 10 min at room temperature, luminescence was read on a GloMax^®^ luminometer. For the determination of IC_50_, IBuDM, IBnDM, BTBnDM, and BFBnDM were tested at 50, 100, 500, and 1000 µM. The results were plotted as dose inhibition curves using nonlinear regression with a variable slope to determine the IC_50_ values of inhibitor compounds (with GraphPad Prism 8.0).

### 2.4. Viability Assay

The cytotoxicity of compounds was assessed using the CCK-8 assay (ab228554; Abcam, Cambridge, UK) on VERO cells. These cells were exposed to serial dilutions of IBuDM, IBnDM, BTBnDM, and BFBnDM for 72 h, followed by incubation with CCK8 tetrazolium salt for 4 h at 37 °C in a CO_2_ incubator. Absorbance was measured at 460 nm using a GloMax^®^ Discover Microplate Reader (Promega, Madison, WI, USA), and the percentage of cellular viability was calculated relative to untreated cells.

### 2.5. Production of SARS-CoV-2 Pseudotyped Particles

To perform a study on the BSL-2 laboratory, the α-SARS-CoV-2 pseudovirus particle production and infection system was constructed using a lentiviral vector bearing luciferase gene reporter for easy observation and analysis. The protocol to generate SARS-CoV-2 pseudoviruses consists of a three-plasmid co-transfection strategy in HEK-293T cells as previously reported [27]. pcDNA3.1(-) SARS-Swt-C9 plasmid for α-SARS-CoV-2 variant was gently provided by Jean K. Millet [28], and o-SARS-CoV-2 variant (BA.2 lineage) plasmid was purchased by Invivogen (Catalog code: p1-spike-v12). Pseudovirus was produced by using Lipofectamine and co-transfection of 300 ng of pCMV-MLVgag-pol, 400 ng of pTG-Luc, and 300 ng of spike encoding vector. During the production of pseudotyped particles, negative and positive controls were produced by replacing the plasmid encoding the spike glycoprotein with an empty vector and VSV-envelope, respectively. The production efficiency of the pseudoviruses was assessed by measuring the luciferase gene expression with the Luciferase Assay System (Promega) according to the manufacturer’s instructions.

### 2.6. In Vitro Pseudovirus Entry Inhibition Assay

For pseudovirus entry inhibition screening, VERO cells were pre-treated with IBuDM, IBnDM, BTBnDM, and BFBnDM for 2 h, followed by transduction with α and o-SARS-CoV-2 spike pseudotyped virus particles. The inhibition of pseudovirus entry was detected by measuring the luminescence values following the addition of luciferin substrate (Luciferase Assay System—Promega). The assay was done in triplicates, and the data were reported as relative light units (RLUs) compared to uninfected control and as a percentage of enzymatic activity inhibition with respect to untreated samples. Calpeptin (2 µM) (Bio-Techne SRL, Milano, Italy, Catalog # 0448) and Camostat (5 µM) were used as a positive control of CatB and TMPRSS2 inhibitors, respectively.

### 2.7. Molecular Docking

AutoDock 4.2.6, supported in YASARA (v. 23.5.19, YASARA Biosciences GmbH, Vienna, Austria), was used for molecular docking studies, by using the crystal structure of SARS-CoV-2 3CL^pro^ (PDB ID: 6LU7), SARS-CoV-2 PL^pro^ (PDB ID: 7LBR), CatB (PDB ID: 2IPP), CatL (PDB ID: 5F02), cathepsin K (CatK) (PDB ID: 5TUN), cathepsin S (CatS) (PDB ID: 4P6E), and cathepsin V (CatV) (PDB ID: 3H6S), obtained from the Protein Data Bank (PDB, http://www.rcsb.org/pdb accessed on 26 April 2023). AutoGrid software (4.2.6) was used to generate the maps, including all surface atoms of the crystallized ligand with a spacing of 0.375 Å. Point charges were initially assigned according to the AMBER03 force field and then damped to mimic the less polar Gasteiger charges used to optimize the AutoDock scoring function. The structure of all ligands was optimized at the semiempirical level of PM6 theory [29]. As previously reported, all parameters were inserted at their default settings [30].

The 2D interactions between the molecule and receptor and the docking consequence, ligand contacts, and surface study were visualized using the BIOVIA Discovery Studio Visualizer 2021 software.

### 2.8. Molecular Dynamics Simulations

The molecular dynamics simulation studies were carried out according to the protocols described in our previous studies [31,32]. The protein–ligand complex was placed in a simple point charge water-modeled periodic cubic box (8 Å) [33].

NaCl ions (0.9%) were added to mimic physiological conditions. An excess of salt was added to neutralize the charge, while water molecules were deleted to readjust the solvent density to 0.997 g/mL.

The AMBER force field was used, and a pressure of 1atm was applied. Simulations were run at 298 K [34,35]. The final dimensions of the box were approximately 80 × 80 × 80 Å3. A short MD simulation was performed on water only to remove interferences. The box was brought to an energy minimum using the steepest descent minimization to remove conformational stress until convergence (<0.01 kcal/mol Å). Each simulation lasted 100 ns, and the ligand–protein contacts that occurred during the MD were recorded every 200 ps. Ligand–protein contacts occurring during MD were analyzed using the function of YASARA software (v. 23.5.19, YASARA Biosciences GmbH, Vienna, Austria).

### 2.9. Statistical Analysis

The results are depicted as the mean ± SD based on a minimum of three independent experiments. Statistical analysis utilized GraphPad Prism 8.0.1.244 software (GraphPad Software Inc., San Diego, CA, USA) and involved one-way analysis of variance (ANOVA). Significance levels of the *p*-values are represented by asterisks (**, ***, ****), indicating values less than 0.01, 0.001, and 0.0001, respectively. The EC_50_ values of inhibitors were calculated from dose inhibition curves using nonlinear regression analysis by GraphPad Prism. The IC_50_ values of inhibitor compounds of Mpro and PLpro proteases were calculated using GraphPad Prism 8.0. from dose inhibition curves using nonlinear regression with a variable slope.

## 3. Results

### 3.1. Docking and Molecular Dynamics Simulations

This study used a pharmacophore model developed in a previously published paper [15]. Molecules with the best binding energy (cutoff −6.5 kcal/mol) were selected. The SMILES of the HIV-1 inhibitors library, in order of decreasing binding energy, were reported in Appendix A. The four selected compounds were also evaluated on PL^pro^.

To rationalize the binding modes of the inhibitors, we docked all compounds into the prepared X-ray crystal structure of 3CL^pro^ (PDB-ID: 6LU7) and PL^pro^ (PDB-ID: 7LBR).

Within the active site of 3CL^pro^, IBuDM established five H-bonds with Leu141, His164, Asp187, Arg188, Thr190, and Gln19. The aliphatic residues in the ligand backbone interacted via hydrophobic interactions with Cys44 and Met49. Two π–sulfur interactions were established with Cys145 (Appendix A).

The docked pose of IBnDM took on a similar conformation (compared to IBuDM) within the active site of 3CL^pro^, achieving the best docking score (Appendix A) among the four selected compounds. In fact, the indole ring established five interactions with the residues of the side chains of the protein: an H bond with Asp187, a π–sulfur interaction with Cys44, π–π interaction with His41, and two hydrophobic contacts with Met49 and Pro52. Additional H-bonds were established with Cys145, Glu166, and Gln189 (Appendix A).

BTBnDM and BFBnDM assumed similar conformations within 3CL^pro^. In fact, both established a H-bond between the sulfonamide group and Glu166. The thiophene ring of BTBnDM was stabilized by π–π interaction and p–sulfur interactions with Met49 and Arg188. A better binding energy value of BFBnDM was probably due to the additional interactions of Cys44, Tyr54, and Arg188 of the benzofuran ring (Appendix A).

For the PL^pro^ SARS-CoV-2 structure, modeling studies demonstrated that all compounds examined interacted with residues in the S3 and S4 subsites. In particular, IBuDM showed a better energy of binding (Appendix A). The main chain and side chains of several residues adapted significantly to accommodate the indole ring of IBuDM within the S3 subsites, establishing a π–π interaction with Gly163 and a π–anion interaction with Asp164, while two H-bonds were established with Leu162 and Tyr268, indole ring nitrogen and methoxy group, respectively (Appendix A). There were two hydrophobic interactions with Arg166 and Tyr268. Additional interactions in the S4 subsites with residues Pro248, Gly268, and Gln269 improve the affinity of the ligand compared to the other compounds.

Compounds IBnDM and BFBnDM had a lower affinity towards PL^pro^. Although both compounds assumed a similar pose within the S3/S4 subsites, IBnDM exhibited a lower binding energy (−9.1 kcal/mol). The H-bond with Leu162 (2.1 Å) and the π–sigma interaction with Gln269 were established with the indole ring (Appendix A). Compared to the other compounds, the π–π interaction with Gly163 is missing. The benzyl ring directly bound to the sulfonamide nitrogen is exposed outside the S4 subsites, establishing a hydrophobic contact with Pro247.

The BTBnDM molecule established numerous contacts in the secondary region compared with the active site. In fact, the thiophene ring is stabilized by interactions with Leu162, Gly163, Asp164, Tyr268, and Gln296 (Appendix A). Further, hydrophobic contributions are derived from residues Arg168, Ala246, and Thr301. All docked poses of the compounds under investigation suggested that the ligands occupy the BL2 loop, blocking access to the active site (Figure 2).

The experimental binding data and the PDB entries for the five cathepsin proteins were reported in Appendix A. The docked pose of IBnDM within the catalytic site of CatB established two H-bonds with Gln23 and Cys26 with the sulfonamide group at a distance of 1.9 Å and 2.0 Å, respectively, and other two H-bonds with Gly198 and Gly74 of the carboxyamide nitrogen and the indole ring, respectively. Additional hydrophobic interactions were shown with Cys26, Pro76, His111, Val176, and Ala200 (Appendix A). Unlike the binding with CatL, in this case, IBnDM was better hosted by the enzyme in the interaction sites. The greater stability of the complex during the MD simulations compared to the other complexes (see Appendix A) was probably due to the greater number of ligand interactions. Furthermore, IBnDM appeared to be more buried within the catalytic site of CatB than CatL (Figure 3). The RMSD of the protein fluctuated by 1 Å at 45 ns and then remained constant. The ligand stabilized after 15 ns, showing slight fluctuations during the simulation. Furthermore, the number of hydrogen bonds in the complex did not show significant variations, demonstrating the stability of the complex (Appendix A).

The main interactions of IBnDM with primary chain residues of the active site of CatL were mediated by hydrogen bonds with Cys25, Asp162, His163, and Ala138 (Appendix A). The indole ring and benzyl fragments establish hydrophobic contacts with Met70, Ala135, Leu144, and a π–π interaction with Cys22. MD simulations were performed to analyze the stability of the CatL/IBnDM complex. The RMSD of the ligand underwent a notable change of 5 Å after approximately 2 ns and subsequently stabilized, while the potential energy of the system remained stable for the entire duration of the simulation. The RMSD of the protein structure showed small fluctuations, reaching stability after 45 ns. The IBnDM likely adapted to the active site pocket before reaching equilibrium within the complex (Appendix A).

The CatK/IBnDM complex showed a similar pose to the previous ones, establishing an H-bond with Asn161 and Met68, while five hydrophobic contacts stabilized the complex, also evident from the RMSD of the ligand throughout the simulation (Appendix A).

The MD simulation of the CatS/IBnDM complex showed a fluctuation in the RMSD of the protein structure. In fact, the RMSD of IBnDM displayed an increase of 2.5 Å after 60 ns. The low stability of this complex was probably due to a lower number of interactions of IBnDM within the active site. Indeed, the ligand appeared to be exposed outward, forming only one H-bond with Gly62 and two p–p interactions with Asn163 and Phe70 (Appendix A).

The CatV/IBnDM complex underwent a sharp change in the RMSD of the ligand after 5 ns, remaining stable for the entire duration of the experiment. Of note, as shown in the docked pose (Appendix A), the ligand formed three H-bonds in all sub-pockets of the active site, leaving a benzyl fragment exposed.

### 3.2. In-Vitro Screening for 3CL^pro^ and PL^pro^ Inhibition

To verify the anti-protease activity of compounds, we monitored the enzymatic activity of 3CL^pro^ and PL^pro^ in vitro in the presence or absence of selected compounds. Recombinant SARS-CoV-2 3CL^pro^ and PL^pro^ were separately combined with the luminogenic 3CL^pro^ and PL^pro^ substrate and the compounds for 60 min at 4 °C. The reaction was then stopped by adding Luciferin Detection Reagent, and the luminescence was recorded on a plate-reading luminometer. The GC376 was used as a positive control. The results, expressed as relative luminescence units (RLU) (Figure 4A) and percentage of enzymatic activity (Figure 4B), reported that a non-toxic concentration of IBuDM, IBnDM, BTBnDM, and BFBnDM inhibits the enzymatic activity of 3CL^pro^. Among these compounds, 80% of the enzymatic activity of 3CL^pro^ was inhibited following the treatment with IBnDM. Otherwise, all compounds reduced the enzymatic activity of PL^pro^ (Figure 5). As expected, GC376, used as a control, fully inhibited 3CL^pro^ in the low nanomolar range and was moderately active against PL^pro^. Moreover, our studies revealed the inhibition of the viral protease PL^pro^ by Camostat, a compound previously accepted as a specific inhibitor of the cellular protease TMPRSS2 [36,37]. Consequently, our findings propose a dual mechanism of action for Camostat, involving the inhibition of both viral PL^pro^ and host TMPRSS2. For the determination of IC_50_, IBuDM, IBnDM, BTBnDM, and BFBnDM were tested at 50, 100, 500, and 1000 µM. The results were plotted as dose inhibition curves using nonlinear regression with a variable slope and reported in Appendix A.

### 3.3. Inhibition of SARS-CoV-2 Spike Pseudotyped Particle Transduction by Using Cathepsin and TMPRSS2 Inhibitors

The entry of SARS-CoV-2 occurs either through direct fusion with cellular surface or endocytic uptake. In both cases, the S protein undergoes cleavage mediated by TMPRSS2 and cathepsins [38]. In this scenario, the host proteases represent rational targets for COVID-19 therapeutic intervention. Considering the results reported on the moderate activity against viral proteases, we evaluated whether the compounds acted against cellular proteases directly involved in the entry. Thus, we used two variants of pseudoviruses that use different internalization mechanisms and tested the inhibitory effect on entry by pre-treating the cells with a non-toxic concentration of substances (Appendix A). The production and characterization of both pseudoviruses were reported in Appendix A. It is worth noting that TMPRSS2 is indispensable for the entry of α-SARS-CoV-2 but not for o-SARS-CoV-2 [39]. Therefore, we incubated VERO cells with a non-toxic concentration of IBuDM, IBnDM, BTBnDM, and BFBnDM for 2 h and then transduced with a- or o-SARS-CoV-2 pseudoviruses for a further 2 h. The selective entry inhibitors of CatB/L (Calpeptin, 2 µM) and TMPRSS2 protease (Camostat, 5 µM) were used as positive controls. The infectivity of the pseudotyped viral particles was measured by luciferase assay 72 h post-transduction. The results reported in Figure 6 show that all compounds can block the entry of both SARS-CoV-2 pseudoviruses into the cells. However, considering that ο-SARS-CoV-2 enters by a TMPRSS2-independent mechanism, as highlighted by the lack of entry inhibitory effect following Camostat treatment, we speculate that the compounds specifically block cathepsin-mediated endocytosis, proposing them as potential inhibitors of cysteine proteases such as cathepsin.

## 4. Discussion

The evolution of the SARS-CoV-2 virus has led to several variants of concern, with diversified transmissibility, severity, and immune evasion capabilities. Despite the planning and organization of the COVID-19 vaccination campaign, the virus’s rapid mutation rate necessitates continuous efforts to develop effective therapeutic strategies. Different studies encouraged the use of drugs approved for medical conditions different from COVID-19 [40]. This strategy can be particularly relevant for a sustainable response against SARS-CoV-2. Indeed, drugs that are already approved have well-documented mechanisms of action, safety profiles, and known pharmacokinetics.

Recently, the FDA approved Paxlovid^TM^ as the first oral drug in the treatment of COVID-19. It is a combination of two drugs, nirmatrelvir and ritonavir, and is developed by Pfizer. Nirmatrelvir is a protease inhibitor with in vitro activity against SARS-CoV-2, while ritonavir is included to boost the levels of nirmatrelvir in the body. The clinical trials indicated that Paxlovid^TM^ significantly reduced the risk of hospitalization and death in individuals with COVID-19 [41]. Paxlovid^TM^ had received Emergency Use Authorization (EUA) from regulatory authorities in various countries and is generally recommended for use in individuals with mild-to-moderate COVID-19 who are at high risk of progressing to severe disease. In the context of infectious diseases like COVID-19, the emergence of new variants or strains is a dynamic challenge. Continuous research and the study of new drugs are essential for adapting and responding effectively to evolving conditions. Investigating the antiviral activity of these new drugs not only provides potential treatment options for existing variants but also ensures preparedness for future mutations or strains that may arise. Repurposing leverages this knowledge, potentially speeding up the drug development process compared to developing entirely new drugs. Developing a new drug from scratch can be a lengthy and costly process. Repurposing existing drugs can significantly reduce the time and cost of bringing a treatment to the market. This scenario is critical in rapidly evolving viral infections such as SARS-CoV-2. Besides, repurposed drugs can be combined with other drugs or therapies to enhance their effectiveness or address multiple aspects of the disease. Based on these considerations, this study proposes in silico and in vitro approaches to identifying and developing potential therapeutic options for SARS-CoV-2 infection. Combining in silico and in vitro approaches allows us to efficiently identify and validate potential drug candidates within a library already validated for HIV-1 for treating SARS-CoV-2 [15,16]. The crystal structure of SARS-CoV-2 3CL^pro^ in complex with the N3 inhibitor was previously solved by high-resolution X-ray crystallography (2.16 Å). In this complex, the N3 peptide is covalently linked to the amino acid Cys145 [42]. IBuDM is the only compound that establishes an H-bond with His164 via the carboxyamide nitrogen. The most promising inhibitor (IBnDM) establishes the crucial hydrogen bond with the imidazole of Cys145 in the S1 pocket of the protease through the hydroxyl group at a distance of 2.0 Å, while Leu27, Met49, and Pro52 establish lipophilic contacts. All substituents of the carboxyamide fragment establish a π–π interaction with His145, settling inside the active site pocket.

The active site of PL^pro^ can generally be divided into four subsites named S1–S4. The active site, located in the S1 and S2 domain, formed by the catalytic triad Cys111, His272, and Asp286, is located in a solvent-exposed cleft at the interface of the thumb and palm domains. The S1 and S2 subsites adopt a narrow, linear tunnel conformation without well-defined binding pockets before opening to the catalytic triad. The binding of host and virus protein substrates is controlled by the flexible b-hairpin BL2 loop, which contains an unusual beta turn formed by Tyr268 and Gln269, which controls access to the active site.

That said, the active site area of PL^pro^ possesses interesting characteristics that challenge structure-based drug design efforts. For example, several regions in the protein’s active site are dynamic, including the BL2 loop in the catalytic site, which can adopt different conformations upon interaction with inhibitors or substrates. In particular, most of the design efforts to date have focused on the S3/S4 subsites opened to address PL^pro^ functionality [43].

The docked poses of the test compounds suggest that the ligands stabilize the BL2 loop, blocking access to the active site used by the virus to contain the PL^pro^ cleavage motif. However, the important aspect of the mechanism, which may be the key to these inhibitors, relies on a flexible segment, BL2. Tyr268, at the tip of the b-hairpin, retains the benzyl groups of IBnDM, BTBnDM, and BFBnDM, burying it almost entirely in the enzyme, while it establishes H-bonds with IBuDM, stabilizing the sulfonamide group. Interestingly, the carboxyamide substituents in all compounds are accommodated inside the pocket under the BL2 loop.

The results reported through in silico approaches were also corroborated by in vitro assays. We reported that non-toxic concentrations of the compounds moderately inhibit the enzymatic activity of 3CL^pro^ and PL^pro^. IBnDM is particularly effective, inhibiting 80% of the enzymatic activity of 3CL^pro^. Unlike this, the compounds effectively block the entry of both α- and ο-SARS-CoV-2 pseudoviruses into the cells. The inhibitory effect is observed even though o-SARS-CoV-2 is entered by a TMPRSS2-independent mechanism. Although the proposed structures showed a mild inhibitory activity against viral proteases, they effectively arrested the entry of both SARS-CoV-2 pseudoviruses into the cells; we hypothesize that the compounds specifically inhibit cathepsin-mediated endocytosis, proposing them as potential inhibitors of cysteine cathepsin proteases. Consequently, we focused computational studies on the cathepsin family (CatB, CatL, CatK, CatS, and CatV). As mentioned, SARS-CoV-2 viral particles enter into the cell via clathrin-mediated endocytosis followed by proteolytic cleavage of the S protein by CatL. Furthermore, other cysteine cathepsins have been reported to be essential for processing the viral S protein upon entry into the cell. Based on the proteolytic mechanism, the cysteine and histidine of the catalytic triad within the active site form a pair of thiolate-imidazolium ions, which confers high nucleophilicity to the cysteine thiol. Under these conditions, the cysteine thiol acts as a nucleophile to hydrolyze the scissile bond of the substrate [44]. The most potent covalent inhibitors of CatL include peptidomimetic compounds with short peptide sequences to resemble structural motifs of the natural substrate, which constitutes a recognition part and a reactive part, an electrophilic moiety known as warhead that promotes cysteine attachment. Noncovalent inhibitors have also been designed to target the cathepsin family [45]; IBnDM showed lower binding energy across the entire cathepsin family. Furthermore, the more significant number of contacts within the CatB active site and sub-pockets, as well as the better stability of the complex during MD simulation, suggests that it could be a potential selective inhibitor of CatB.

The best values for the four compounds have an overall trend in favor of CatB and CatS. This detail could be linked to the intrinsic nature of the inhibitors and the structure–activity relationship. From a drug design perspective, a greater affinity against cathepsins could be achieved through structural optimization of IBnDM. According to our computational investigations, structural modifications of IBnDM could allow interactions with the S1 pocket of CatL. However, cysteine cathepsins may play redundant roles in viral entry that are consistent with inhibiting several cathepsins. This suggests that the strict selectivity of inhibitors for individual cathepsins is not the desired property in viral infections.

## 5. Conclusions

In the current study, an in-silico strategy was employed to explore the potential of HIV-1 protease inhibitors, previously synthesized by our group, in fighting COVID-19 [25,26].

The computational approach employed the same in silico model proposed in our previous study [23]. We reported that non-toxic concentrations of the compounds moderately inhibit the enzymatic activity of 3CL^pro^ and PL^pro^. IBnDM is particularly effective, inhibiting 80% of the enzymatic activity of 3CL^pro^. Interestingly, in vitro tests showed that the compounds effectively block the entry of both SARS-CoV-2 pseudoviruses into the cells. We hypothesized that the tested drugs inhibit the activity of cathepsins. Computational studies have highlighted the key interactions between the inhibitors (IBuDM, IBnDM, BTBnDM, and BFBnDM) and the active site of cathepsins, proposing them as potential noncovalent cathepsin inhibitors.

This study hypothesizes that the examined compounds show potential inhibitory effects on the entry of SARS-CoV-2 pseudoviruses into the cells, potentially through the targeting of cathepsin-mediated endocytosis. This information could contribute to the development of therapeutic strategies against COVID-19.

## Figures and Tables

**Figure 1 viruses-16-00338-f001:**
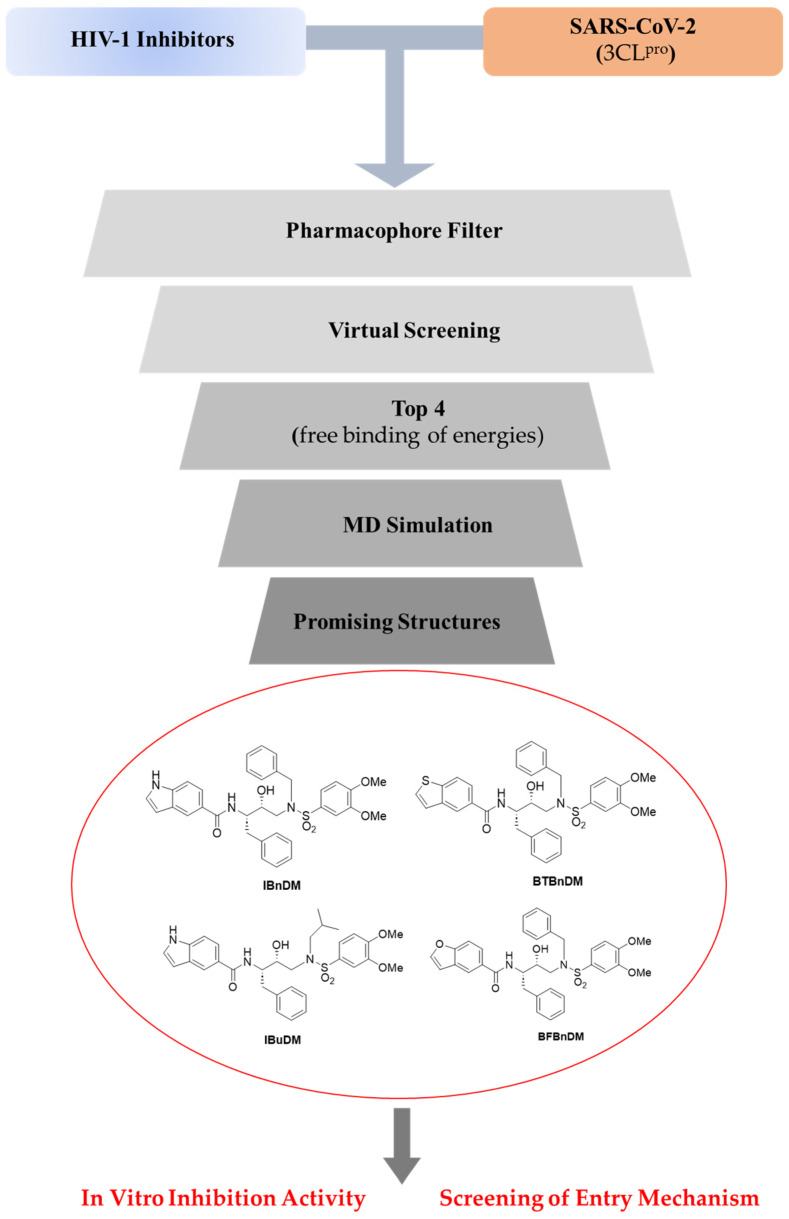
Workflow of the integrated experimental approach. Compounds were selected, and an integrated strategy of in silico and in vitro assays was reported against viral protease and spike protein of SARS-CoV-2.

**Figure 2 viruses-16-00338-f002:**
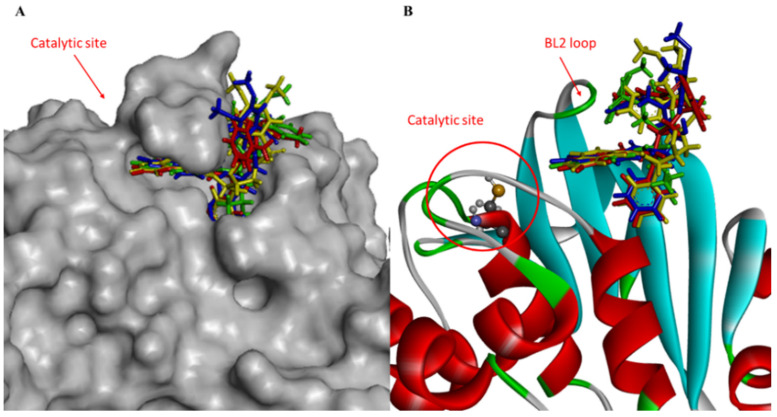
(**A**) Close-up view of hydrophobic surface (gray) of the active site of PL^pro^ in complex with IBuDM (green), IBnDM (red), BTBnDM (yellow), and BFBnDM (blue) within a sub-pocket; (**B**) and close-up view of ribbon model.

**Figure 3 viruses-16-00338-f003:**
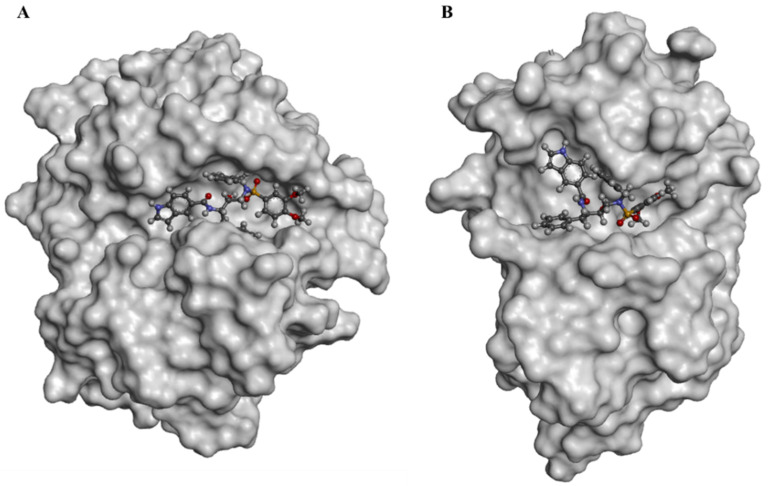
Surface representation of docked poses of IBnDM within the active site of CatB (**A**) and CatL (**B**).

**Figure 4 viruses-16-00338-f004:**
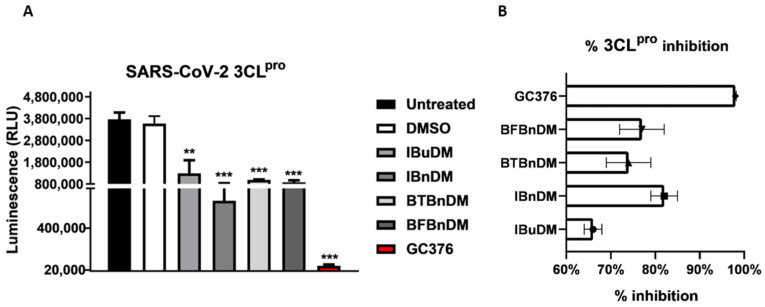
In vitro inhibition of 3CL^pro^ activity. Purified recombinant SARS-CoV-2 3CL^pro^ was combined with 500 µM of IBuDM, or IBnDM, BTBnDM, and BFBnDM, and the 3CL^pro^ inhibitor GC376 (2 µM) diluted in assay buffer (50 mM HEPES pH 7.2, 10 mM DTT, and 0.1 mM EDTA) and added to 3CL^pro^ substrate solution (40 µM) in an opaque white 96-well plate for 60 min at 37 °C. DMSO was included as a control and used at 0.5%. Then, reactions were terminated by adding 50 µL of Luciferin Detection Reagent, and after 20 min at room temperature, luminescence was recorded on a GloMax^®^ luminometer. The results were expressed as relative luminescence units (RLU) (**A**) and percentage of enzymatic activity inhibition (**B**) and represent the mean ± SD based on a minimum of three independent experiments. Statistical analysis utilized GraphPad Prism 8 software (GraphPad Software, San Diego, CA, USA) and involved one-way analysis of variance (ANOVA). Asterisks represent significance levels of the *p*-values. ** and *** indicate values less than 0.01 and 0.001, respectively.

**Figure 5 viruses-16-00338-f005:**
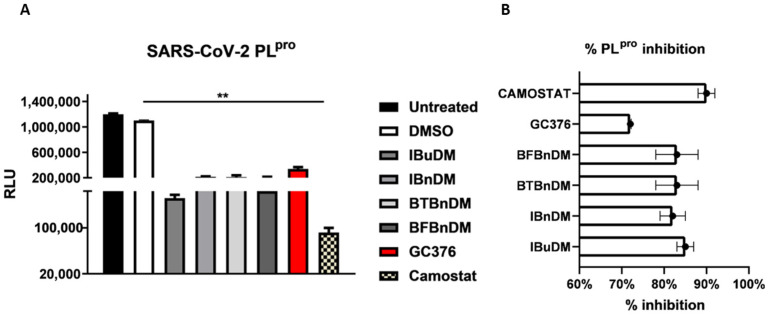
In vitro inhibition of PL^pro^ activity. Purified recombinant SARS-CoV-2 PL^pro^ was combined with 500 µM of IBuDM, IBnDM, BTBnDM, and BFBnDM, and the 3CL^pro^ inhibitor GC376 (2 µM) and Camostat (5 µM), diluted in assay buffer (50 mM HEPES pH 7.2, 10 mM DTT, and 0.1 mM EDTA) and added to 3CL^pro^ substrate solution (40 µM) in an opaque white 96-well plate for 60 min at 37 °C. DMSO was included as a control and used at 0.5%. Then, reactions were terminated by adding 50 µL of Luciferin Detection Reagent, and after 20 min at room temperature, luminescence was recorded on a GloMax^®^ luminometer. The results were expressed as relative luminescence units (RLU) (**A**) and percentage of enzymatic activity inhibition (**B**) and represent the mean ± SD based on a minimum of three independent experiments. Statistical analysis utilized GraphPad Prism 8 software (GraphPad Software, San Diego, CA, USA) and involved one-way analysis of variance (ANOVA). Asterisks represent significance levels of the *p*-values. ** indicates a value less than 0.01.

**Figure 6 viruses-16-00338-f006:**
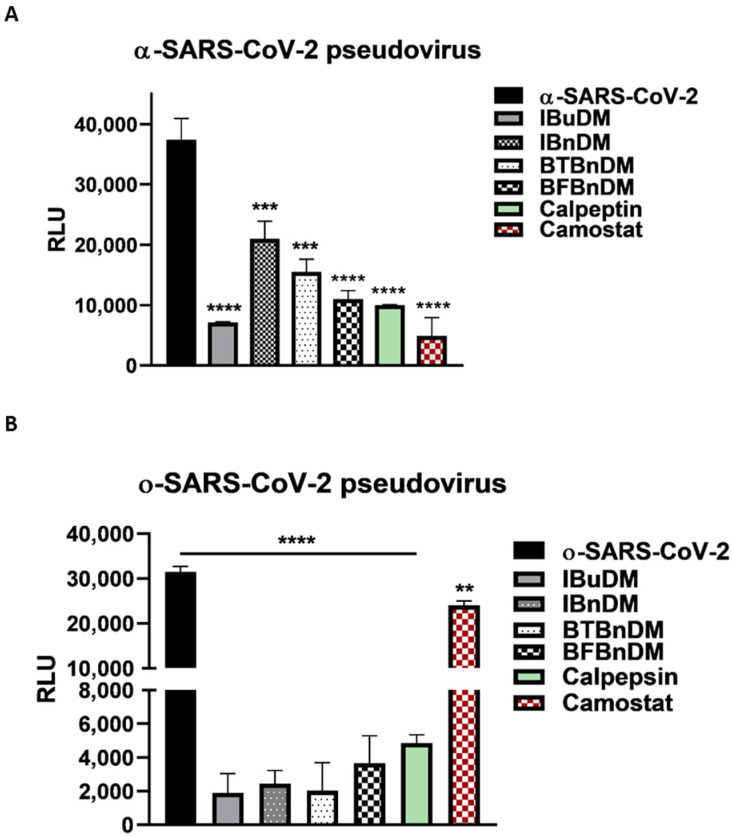
SARS-CoV-2 spike pseudotyped particle inhibition assay. Internalization of alpha (**A**) and omicron (**B**) SARS-CoV-2 pseudoviruses into Vero cells, following pre-treatment of the cells with IBuDM (100 µM), IBnDM (50 µM), BTBnDM (100 µM) BFBnDM (100 µM), Camostat (5 µM, inhibitor of the serine protease TMPRSS2), and Calpeptin (2 µM, CatB/L) for 1 h at 37 °C. The results are the means ± SD of triplicate analyses and are expressed as relative luminescence units (RLU). Statistical analysis utilized GraphPad Prism 8 software (GraphPad Software, San Diego, CA, USA) and involved one-way analysis of variance (ANOVA). Significance levels of the *p*-values are represented by asterisks (**, ***, ****), indicating values less than 0.01, 0.001, and 0.0001, respectively.

## Data Availability

The original contributions presented in the study are included in the article/Appendix A, further inquiries can be directed to the corresponding author/s.

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
