# Peer review of "Targeting Viral and Cellular Cysteine Proteases for Treatment of New Variants of SARS-CoV-2"

_viruses, 2024, doi:10.3390/v16030338_

Round 1
Reviewer 1 Report
Comments and Suggestions for Authors
The article "Targeting viral and cellular cysteine proteases for treatment of new variants of SARS-CoV-2"The study proposes the use of HIV-1 protease inhibitors against SARS-CoV-2. The group already has other studies in the area.
17.Funicello, M.; Chiummiento, L.; Tramutola, F.; Armentano, M.F.; Bisaccia, F.; Miglionico, R.; Milella, L.; Benedetti, F.; Berti, F.; 571
Lupattelli, P. Synthesis and Biological Evaluation in Vitro and in Mammalian Cells of New Heteroaryl Carboxyamides as HIV-
Protease Inhibitors. Bioorg Med Chem 2017, 25, 4715–4722, doi:10.1016/j.bmc.2017.07.014. 573
18. D’Orsi, R.; Funicello, M.; Laurita, T.; Lupattelli, P.; Berti, F.; Chiummiento, L. The Pseudo-Symmetric N-Benzyl Hydroxyethyl- 574 amine Core in a New Series of Heteroarylcarboxyamide HIV-1 Pr Inhibitors: Synthesis, Molecular Modeling and Biological Eval- 575
uation. Biomolecules 2021, 11, 1584, doi:10.3390/biom11111584 With the rapid evolution of SARS=COV-2, repositioned drugs or be combined with other drugs or therapies to increase their efficacy or address multiple aspects of the disease.
Based on these considerations, this study proposes in silico and in vitro approaches to identify and develop potential therapeutic options for SARS-CoV-2 infection. The combination of in silico and in vitro approaches could make it possible to identify and validate potential drug candidates in a library validated for HIV-1 for the treatment of SARS-CoV-2
Author Response
We are grateful to the reviewer the positive comments. Nevertheless, we are resubmitting a revised version of the manuscript.
Reviewer 2 Report
Comments and Suggestions for Authors
The manuscript, entitled “Targeting viral and cellular cysteine proteases for treatment of new variants of SARS-CoV-2” deals with in silico and in vitro studies targeting cystein proteases. The manuscript is well written , and interesting for the readers with very good results for some compounds. Nevertheless I have one question where authors performed validation test for docking studies. Furthermore, 2D diagrams of docking have very low resolution.Also there are some minor comments below.
Line 55 in vitro should be italics
Line 65 Please specify TMPRSS2
Author Response
We sincerely thank the reviewer for the comments, which helped us in improving the revised version of the manuscript. Please find below a point-by-point description that includes the original reviewer's comments in boldface and the responses in red regular typeface. A revised version of the manuscript has been resubmitted.
The manuscript, entitled “Targeting viral and cellular cysteine proteases for treatment of new variants of SARS-CoV-2” deals with in silico and in vitro studies targeting cystein proteases. The manuscript is well written , and interesting for the readers with very good results for some compounds.
- Nevertheless I have one question where authors performed validation test for docking studies.
The docking validation was performed on the 3CLpro protease as published in the previous work [https://doi.org/10.3390/md18040225]. In that case, the authors used a pharmacophore model, built by the Pharmit server ( http://pharmitcsb.pitt.edu/ ) starting from SARS-CoV-2 3CLpro (PDB ID: 6LU7) and with the complexed structure of the ligand N3 (PRD_002214). In the present study, the docking validation was performed on the co-crystallized ligand.
- Furthermore, 2D diagrams of docking have very low resolution.
We modified the 2D diagrams of docking accordingly. Line 55 in vitro should be italics
We thank the reviewer for the suggestion. We edited the manuscript accordingly.
- Line 65 Please specify TMPRSS2
We thank the reviewer for the suggestion. We edited the manuscript accordingly.
Reviewer 3 Report
Comments and Suggestions for Authors
Gentile et al submitted a manuscript titled "Targeting viral and cellular cysteine proteases for treatment of new variants of SARS-CoV-2" for publication in Viruses.
The work is fine and written well. However, the authors need to address few minor details before it can be accepted for publication.
Fig. 1
Remove structures of drugs. It is not your finding, so it is not only not required but also inappropriate. Simply cite the previous work detailing the structures if you feel that their presence is significant in the paper.
Figs. 2 & 3.
In figure description, always mention the program used to generate ribbon/surface projections. Also describe all components presented in the figures, here and elsewhere.
Figs. 4, 5 and 6.
In figure descriptions, mention the tests used to calculate significance. In methods, you don't have to mention if you did, but it is a must in figure descriptions.
Author Response
We sincerely thank the reviewer for the comments, which helped us in improving the revised version of the manuscript. Please find below a point-by-point description that includes the original reviewer's comments in boldface and the responses in red regular typeface. A revised version of the manuscript has been resubmitted.
Gentile et al submitted a manuscript titled "Targeting viral and cellular cysteine proteases for treatment of new variants of SARS-CoV-2" for publication in Viruses. The work is fine and written well. However, the authors need to address few minor details before it can be accepted for publication.
- 1: Remove structures of drugs. It is not your finding, so it is not only not required but also inappropriate. Simply cite the previous work detailing the structures if you feel that their presence is significant in the paper.
We thank the reviewer for the suggestion. However, we believe that it is important to keep the structure of the drugs in Figure 1. The reader has an immediate view of the chemical structures selected by the workflow model used by the authors, arousing more interest. Moreover, the results about the interactions between the moieties of drugs and the enzymes in the docking and molecular dynamics simulation become more evident.
- 2 & 3: In figure description, always mention the program used to generate ribbon/surface projections. Also describe all components presented in the figures, here and elsewhere.
We thank the reviewer for the observation. We mentioned the program used to generate ribbon/surface projections, in the Molecular Docking section.
Figs. 4, 5 and 6: In figure descriptions, mention the tests used to calculate significance. In methods, you don't have to mention if you did, but it is a must in figure descriptions.
We thank the reviewer for the observation. We reported the description of the statistical analysis in the figure legend.
Reviewer 4 Report
Comments and Suggestions for Authors
In this extensive study, the possibility is explored of repurposing HIV-1 protease inhibitors as antiviral drugs for the treatment of SARS-CoV-2 variants. As an initial approach, using a model developed in a previous study, a library of HIV-1 proteases was screened in silico for their ability to dock into the active site pockets of two different viral proteases, 3CLpro and PLpro. Four anti-HIV compounds were able to interact and were subsequently shown to block the entry of the two viral variants in a pseudotype-based assay. However, since the two variants differ in their dependence on the TMPRSS2 cellular protease for entry, rather than targeting the above proteases, the authors conclude that the compounds act by specifically blocking cathepsin-mediated endocytosis. To their credit, the authors do evaluate the ability of one of the potential inhibitors to dock into the active sites of CatB and CatL.
While the in silico and pseudotype-based data are impressive and the studies are extremely well done, the authors really can only speculate as to the activity of these inhibitors against cysteine cathepsin proteases. Without actually confirming their inhibitory activity against this class of proteases, it is premature to go any further than that. Thus, the authors need to tone down a bit their conclusions from these studies, most importantly in the title. Actually, is it not reasonable to ask the authors to test at least one of the inhibitors in in vitro assays with these enzymes?
The authors use the terms a-SARS-Cov-2 and o-SARS-Cov-2, which I assume refer to the alpha and omicron variants. This needs to be specified and elaborated upon.
Comments on the Quality of English Language
Requires significant attention.
Author Response
We sincerely thank the reviewer for the comments, which helped us in improving the revised version of the manuscript. Please find below a point-by-point description that includes the original reviewer's comments in boldface and the responses in red regular typeface. A revised version of the manuscript has been resubmitted.
In this extensive study, the possibility is explored of repurposing HIV-1 protease inhibitors as antiviral drugs for the treatment of SARS-CoV-2 variants. As an initial approach, using a model developed in a previous study, a library of HIV-1 proteases was screened in silico for their ability to dock into the active site pockets of two different viral proteases, 3CLpro and PLpro. Four anti-HIV compounds were able to interact and were subsequently shown to block the entry of the two viral variants in a pseudotype-based assay. However, since the two variants differ in their dependence on the TMPRSS2 cellular protease for entry, rather than targeting the above proteases, the authors conclude that the compounds act by specifically blocking cathepsin-mediated endocytosis. To their credit, the authors do evaluate the ability of one of the potential inhibitors to dock into the active sites of CatB and CatL.
- While the in silicoand pseudotype-based data are impressive and the studies are extremely well done, the authors really can only speculate as to the activity of these inhibitors against cysteine cathepsin proteases. Without actually confirming their inhibitory activity against this class of proteases, it is premature to go any further than that. Thus, the authors need to tone down a bit their conclusions from these studies, most importantly in the title. Actually, is it not reasonable to ask the authors to test at least one of the inhibitors in in vitro assays with these enzymes?
We understand the referee concerns.
In the new version of the manuscript, we state that we hypothesize that the inhibitory activity of the compounds is directed against cathepsin. While we currently lack the immediate capability to incorporate these data, our conclusions highlight the intention to conduct further in vitro tests to confirm that the compounds directly inhibit the enzymatic activity of cathepsin. Although we do not havethe immediate opportunity to add this data, it will be considered in subsequent analyses.
- The authors use the terms a-SARS-Cov-2 and o-SARS-Cov-2, which I assume refer to the alpha and omicron variants. This needs to be specified and elaborated upon.
We regret the oversight. We have included and outlined the characteristics of the two variants, alpha and omicron, in the abstract where they are initially introduced. Additionally, we have rectified the symbols used in the text.
- Comments on the Quality of English Language: Requires significant attention.
We express our appreciation to the reviewer for its valuable suggestion. The manuscript underwent meticulous analysis by a skilled linguist, and we trust that the extensive editing has enhanced the paper's readability.
Reviewer 5 Report
Comments and Suggestions for Authors
This paper by Gentile et al follows previous attemts in the literature (not cited) to utilize HIV protease inhibitors against SARS CoV-2 Proteases. The idea is not new and previous studies already implied that due to the different mechanisms it is not likely that the HIV PR inhibitors would be sufficiently inhibited by the therapeutically relevant transition state mimic inhibitors, and alternative targets were also suggested (e.g. Mahdi et al., PMID: 33243253).
It appears that there are serious problems with the manuscript, regarding the in vitro assays. The authors use luminometric assay to do the in vitro inhibition studies. The give a percentage inhibition at a given concentration, no Ki or IC50 values were determined. The authors state that IBnDM causes 80% inhibition of 3CLpro activity, but according to Figure 4, this inhibition was achieved by using the inhibitor in 500 mM final concentration (based on the molecular weight and stock concentration (16 mg/ml) of 3CLpro, the 3CLpro had 0.45-0.50 mM concentration, although, this is not indicated in the manuscript). In figure 5, PLpro inhibition assay was also performed by using the inhibitor in 500 mM concentration. This concentration can not be considered to be therapeutically relevant. Furthermore, in this high concentration any material would trigger influences on the measurements (like changing pH, etc) so without showing the cleavage inhibition these values are meaningless. As it turned out that these inhibitors do not really work on the viral proteases but might inhibit the cellular proteases required for entry, the Ki or IC50 measurement for those enzymes would be necessary.
It is stated that the aim of the study is to “repurpose” HIV-1 protease inhibitors against SARS-CoV-2 (e.g. in the abstract). But, the use of “repurpose” phrase in this context is misleading because the investigated inhibitor candidates were synthesized previously by the authors (see e.g. “Synthesis and biological evaluation in vitro and in mammalian cells of new heteroaryl carboxyamides as HIV-protease inhibitor” paper) but none of them has been approved by FDA as a therapeutic HIV-1 protease inhibitor.
Minor points:
- Authors did not discuss that a reversible covalent inhibitor against SARS-CoV-2 main protease (nirmatrelvir) is available and a drug containing this inhibitor (Paxlovid) has already been accepted for the treatment of COVID-19 patients.
- It is not discussed in the conclusions sections whether the HIV-1 protease inhibitor candidates - that were tested in silico and in vitro, as well - have a potential to inhibit SARS-CoV-2 protease. (They do not)
- Based on the figures that show the results of in vitro inhibition assays (Figure 4 and 5, respectively), “untreated” samples were used as controls, although, all the compounds were solubilized in DMSO (see section 2.2 in Materials and methods). It must be discussed whether the DMSO as solvent interferes with the protease activity and the corresponding results must also be shown.
- - The authors state that they found inhibition of TMPRSS2 by camostat (Line 361-362), but the corresponding results are not presented or cited.
Author Response
We sincerely thank the Reviewer for the comments, which helped us improve the revised version of the manuscript. Please find below a point-by-point description that includes the original reviewer's comments in boldface and the responses in regular typeface. A revised version of the manuscript has been resubmitted.
Comments and Suggestions for Authors
This paper by Gentile et al follows previous attemts in the literature (not cited) to utilize HIV protease inhibitors against SARS CoV-2 Proteases. The idea is not new and previous studies already implied that due to the different mechanisms it is not likely that the HIV PR inhibitors would be sufficiently inhibited by the therapeutically relevant transition state mimic inhibitors, and alternative targets were also suggested (e.g. Mahdi et al., PMID: 33243253).
We appreciate your observations regarding our article. We added the following sentence to the manuscript and the references as follow:
" Besides, it has been investigated the efficacy of HIV protease inhibitors (PIs), lopinavir and ritonavir, against SARS-CoV-2 [6], considering their documented activity against related coronaviruses [7,8]. The study focused on the main protease Mpro of SARS-CoV-2 as a po-tential target for these PIs. While in silico screening identified nelfinavir as a potential Mpro inhibitor [9], lopinavir and ritonavir were suggested by molecular dynamics simulation [10]. Previous studies showed the effectiveness of lopinavir/ritonavir against SARS-associated coronavirus [7,8], but recent clinical trials for COVID-19 yielded no sig-nificant benefits [11–13]. Mahdi and collaborators tested a panel of HIV PIs against SARS-CoV-2 Mpro using a cell culture-based model and determined IC50 values. The com-bination of lopinavir plus ritonavir showed the lowest IC50, although with cellular viabil-ity concerns. Darunavir and atazanavir required higher concentrations but exhibited no cytotoxicity. The study suggested limited clinical potential for HIV PIs in treating COVID-19, raising the possibility of other molecular targets for these drugs."
Therefore, we want to address the reviewer's concerns and provide additional clarification on our study. The authors mentioned demonstrated that FDA-approved HIV inhibitors cannot be used as SARS-CoV-2 protease inhibitors for different reasons. The authors underline the fact that additional targets are involved. Although not using FDA-approved inhibitors, our experimental data verifies that the effect on viral proteases is insignificant; instead, the activity is directed against cellular targets that inhibit viral entry. We thought that the inhibitory activity may be on cathepsins. Thus, we add a further effort in the search for anti-SARS-CoV-2 drugs.
It appears that there are serious problems with the manuscript, regarding the in vitro assays. The authors use luminometric assay to do the in vitro inhibition studies. The give a percentage inhibition at a given concentration, no Ki or IC50 values were determined. The authors state that IBnDM causes 80% inhibition of 3CLpro activity, but according to Figure 4, this inhibition was achieved by using the inhibitor in 500 mM final concentration (based on the molecular weight and stock concentration (16 mg/ml) of 3CLpro, the 3CLpro had 0.45-0.50 mM concentration, although, this is not indicated in the manuscript). In figure 5, PLpro inhibition assay was also performed by using the inhibitor in 500 mM concentration. This concentration cannot be considered to be therapeutically relevant. Furthermore, in this high concentration any material would trigger influences on the measurements (like changing pH, etc) so without showing the cleavage inhibition these values are meaningless. As it turned out that these inhibitors do not really work on the viral proteases but might inhibit the cellular proteases required for entry, the Ki or IC50 measurement for those enzymes would be necessary.
We thank the Reviewer for the observation. We apologize for the oversight; regrettably, there was a typing error, and "micromolar" was incorrectly replaced with "millimolar." In addition, we verified that the camostat concentration used as a control was tested at the work solution of 5 micromolar. The concentration of 100 micromolars was referred to as the original stock solution. We apologize for the error and have corrected the concentrations in the revised manuscript.
It is stated that the aim of the study is to “repurpose” HIV-1 protease inhibitors against SARS-CoV-2 (e.g. in the abstract). But the use of “repurpose” phrase in this context is misleading because the investigated inhibitor candidates were synthesized previously by the authors (see e.g. “Synthesis and biological evaluation in vitro and in mammalian cells of new heteroaryl carboxyamides as HIV-protease inhibitor” paper) but none of them has been approved by FDA as a therapeutic HIV-1 protease inhibitor.
We appreciate the Reviewer's feedback. The manuscript has been revised, and the term "repurposing" has been excluded from the aim of the study. Nevertheless, based on the in vitro results obtained through pseudovirus transduction, we have retained the notion that the compounds show potential utility as repurposed drugs for future evaluations. The concept of investigating the potential of drugs that the FDA still needs to approve stems from the recognition that existing treatments may not always be sufficient, especially in the context of emerging diseases or conditions like the COVID-19 pandemic. When there is an urgent need for effective treatments, and no approved options are available, exploring unapproved drugs becomes a critical avenue. We believe that the results of each study contribute valuable data to the scientific community. In the case of infectious diseases like COVID-19, new variants or strains may emerge over time. Investigating the potential of various drugs, even those not yet approved allows for a more adaptable response to evolving conditions.
Minor points:
- Authors did not discuss that a reversible covalent inhibitor against SARS-CoV-2 main protease (nirmatrelvir) is available and a drug containing this inhibitor (Paxlovid) has already been accepted for the treatment of COVID-19 patients.
We thank the Reviewer for the suggestion. We enriched the discussion section with some considerations about PAXLOVID and added the reference accordingly. We reported this sentence:" Recently, FDA approved the PaxlovidTM as a first oral drugs in the treatment of COVID-19. It is a combination of two drugs, nirmatrelvir and ritonavir, and is developed by Pfizer. Nirmatrelvir is a protease inhibitor with in-vitro activity against SARS-CoV-2, while ritonavir is included to boost the levels of nirmatrelvir in the body. The clinical trials indicated that PAXLOVID significantly reduced the risk of hospitalization and death in individuals with COVID-19 [40]. PAXLOVID had received Emergency Use Authorization (EUA) from regulatory authorities in various countries. PAXLOVID is generally recommended for use in individuals with mild to moderate COVID-19 who are at high risk of progressing to severe disease. Emerging new variants or strains in infectious diseases like COVID-19 is a dynamic challenge. Continuous research and studying new drugs are essential for adapting and responding effectively to evolving conditions. Investigating the antiviral activity of these new drugs not only provides potential treatment options for existing variants but also ensures preparedness for future mutations or strains that may arise."
- It is not discussed in the conclusions sections whether the HIV-1 protease inhibitor candidates - that were tested in silico and in vitro, as well - have a potential to inhibit SARS-CoV-2 protease. (They do not)
We thank the Reviewer for the observation; we added the following sentence to the conclusion:" We reported that no toxic concentrations of the compounds moderately inhibit the enzymatic activity of 3CLpro and PLpro. IBnDM is particularly effective, inhibiting 80% of the enzymatic activity of 3CLpro."
- Based on the figures that show the results of in vitro inhibition assays (Figure 4 and 5, respectively), “untreated” samples were used as controls, although, all the compounds were solubilized in DMSO (see section 2.2 in Materials and methods). It must be discussed whether the DMSO as solvent interferes with the protease activity and the corresponding results must also be shown.
We appreciate the Reviewer's suggestion. Therefore, we verified the enzymatic activity of 3CLpro and PLpro following incubation with DMSO. We reported the results in the text and changed the figure accordingly. The results report that the DMSO does not interfere with the protease's activity. Besides, our experiments involve a concentration of DMSO that is notably lower than that investigated by Rimanshee Arya et al. (https://doi.org/10.1007/s12033-021-00383-y) who verified the impact of DMSO on the enzymatic activity of the PLpro protease. Arya's study involved measuring enzyme activity in the presence of varying concentrations of DMSO, ranging from 1% to 10%. In line with these findings, we acknowledge no significant alteration in activity up to 1% DMSO. However, Arya observed a decline in catalytic activity beyond 1% DMSO, concentrations that we deliberately avoid in our cell cultures due to their recognized toxicity. In a parallel investigation, Nguyen conducted experiments to assess the impact of DMSO on the activity of 3CLpro. Nguyen tested 3CLpro activity across a range of DMSO concentrations, from 0 to 50%. The findings revealed that 3CLpro activity remained relatively unaffected up to 10% (v/v) DMSO. However, a decline in enzymatic activity became evident as the concentration of DMSO surpassed this threshold. At the highest concentration tested, 50% (v/v) DMSO, 3CLpro activity was no longer detectable (https://doi.org/10.3390/molecules26071924).
- The authors state that they found inhibition of TMPRSS2 by camostat (Line 361-362), but the corresponding results are not presented or cited.
We appreciate the Reviewer's careful examination of our manuscript. Regarding the inhibition of TMPRSS2 by camostat, bibliographic data support camostat as a specific inhibitor of TMPRSS2, and therefore, we have included the relevant references 35 and 36 to support this concept. It is essential to highlight that our study's novelty lies in demonstrating camostat's inhibition against PLpro, showing specificity against a viral protease. For better readability of the manuscript, we reformulate the phrase as follows: "Moreover, our studies revealed the inhibition of the viral protease PLpro by Camostat, a compound previously accepted as a specific inhibitor of the cellular protease TMPRSS2 [35,36]. Consequently, our findings propose a dual mechanism of action for Camostat, involving the inhibition of both viral PLpro and cellular TMPRSS2 proteases."
Round 2
Reviewer 4 Report
Comments and Suggestions for Authors
In this revision of a previously reviewed manuscript, the authors present a set of in silico and pseudotype-based studies to assess the possibility of repurposing HIV-1 protease inhibitors for their ability to also block the activities of two different SARS-CoV-2 proteases. Based on their findings, they can conclude that a few of these inhibitors do carry potential. But their conclusion that they act on specific cysteine cathepsin proteases needed to be backed off on.
In the absence of direct studies with these enzymes, this can only be considered speculation.
In this revised version, the authors have converted their conclusions to a more speculative tone, in the process making the significance of the study more commensurate with their findings.
Comments on the Quality of English Language
English usage is better but still requires attention.
Author Response
We are grateful to the Reviewer for the positive comments. Regardless, we are resubmitting a revised version of the manuscript in high-quality English.
Reviewer 5 Report
Comments and Suggestions for Authors
Although this manuscript is a substantially improved version, no attemt was made to proper description of the in vitro inhibition experiments.
Decription of the luminescent assays: (lines 151-161) it is completely useless if somebody would try to repeat the experiment. The volume of the enzyme is not given, so it states mixing of enzyme concentrations (one is given in ug/ml, the other in nM) with 12.5 uL of buffer. Final reaction volume is not stated either only the volume of the detection reagent (50 uL).
Figure 4: It would be absolutely required to determine minimum IC50 values for the inhibitors. Actually 500 uM final inhibitor concentration is still very high and it provided only 60 - 80 % inhibition of 3CLpro
The cytotoxicity data are also very puzzling. For example, based on Figure S8 there is no effect of BFBnDM on cellular proliferation till 100 uM, but the CC50 value for this compound is 1 uM in Table S4. The very same compound caused less than 80% inhibition of 3CLpro so IC50 could be 2-300 uM, therefore it would be in the cytotoxic range.
Author Response
We sincerely thank the reviewer for the comments, which helped us in improving the revised version of the manuscript. Please find below a point-by-point description that includes the original reviewer's comments in boldface and the responses in red regular typeface. A revised version of the manuscript has been resubmitted.
Although this manuscript is a substantially improved version, no attemt was made to proper description of the in vitro inhibition experiments.
Decription of the luminescent assays: (lines 151-161) it is completely useless if somebody would try to repeat the experiment. The volume of the enzyme is not given, so it states mixing of enzyme concentrations (one is given in ug/ml, the other in nM) with 12.5 uL of buffer. Final reaction volume is not stated either only the volume of the detection reagent (50 uL).
We thank the reviewer for the observation. We reported the details of the protocol in material and methods as follows: “The inhibitory activities of the IBuDM, IBnDM, BTBnDM and BFBnDM against 3CLpro and PLpro were determined using the SARS-CoV-2 3CLpro and PLpro Luminescent Assays as reported by manufacturer instructions (CS331201 - SARS-CoV-2 3CLpro and PLpro Luminescent Assays, Promega). Purified recombinant SARS CoV-2 3CLpro (9000 µg/mL) (BPS Bioscience cat# 100823) and purified GST-PLpro (1000 nM) (R&D Systems, Cat# E-611-050) were used. The compounds were tested at 500 µM, GC376 was used as a positive control at 2 µM and DMSO was included as a solvent control at 0.5 %. The assay was performed in opaque white 96 well plates with a final volume of 50 μl. Briefly, 12.5 µL of 3CLpro (a final concentration of 16 µg/mL) was incubated for 60 minutes at 37 °C with 12.5 µL of assay buffer (50 mM HEPES pH 7.2, 10 mM DTT, and 0.1 mM EDTA) containing compounds at final concentration of 500 µM. In the untreated sample, 12.5 µL of 3CLpro was incubated with 12.5 µL of assay buffer. 25 µL of substrate was then added at a final concentration of 40 μM. Reactions were then blocked by adding 50 µL of Luciferin Detection Reagent (Promega cat# V8920), and after 20 minutes at room temperature, luminescence was read on a GloMax® luminometer. Similarly, 12.5 µL of PLpro (a final concentration of 10 nM) was incubated for 30 minutes at 25 °C with 12.5 µL of compounds in the assay buffer. The substrate was then added at a final concentration of 40 μM. Reactions were blocked by adding 50 µL of Luciferin Detection Reagent (Promega cat# V8920), and after 10 minutes at room temperature, luminescence was read on a GloMax® luminometer. For the determination of IC50, IBuDM, IBnDM, BTBnDM and BFBnDM were tested at 50, 100, 500 and 1000 µM. The results were plotted as dose inhibition curves using nonlinear regression with a variable slope to determine the IC50 values of inhibitor compounds (with GraphPad Prism 8.0).”
Figure 4: It would be absolutely required to determine minimum IC50 values for the inhibitors. Actually 500 uM final inhibitor concentration is still very high and it provided only 60 - 80 % inhibition of 3CLpro
We thank the reviewer for the observation. We calculated the IC50 values and added this information in the 3.2 section of the manuscript: “For the determination of IC50, IBuDM, IBnDM, BTBnDM and BFBnDM were tested at 50, 100, 500 and 1000 µM. The results were plotted as dose inhibition curves using nonlinear regression with a variable slope to determine the IC50 values of inhibitor compounds (with GraphPad Prism 8.0) and reported in Table S4”. The values were reported in Table S4 of the Supplementary file.
The cytotoxicity data are also very puzzling. For example, based on Figure S8 there is no effect of BFBnDM on cellular proliferation till 100 uM, but the CC50 value for this compound is 1 uM in Table S4. The very same compound caused less than 80% inhibition of 3CLpro so IC50 could be 2-300 uM, therefore it would be in the cytotoxic range.
We sincerely apologize for any inconvenience caused by the formatting issues encountered in the documents, especially regarding the insertion of symbols. We truly appreciate the careful analysis you have conducted. The values provided in Table S5 are indeed in millimolar concentration, indicating that the concentrations used are not toxic. We regret any confusion caused by the incorrect symbol used and promptly rectified the error.
Round 3
Reviewer 5 Report
Comments and Suggestions for Authors
The authors added the required parts to the Materials and Methods and apparently performed the IC50 determinations. These data appear only in the supplementary material as the authors did not want to implement large changes to the manuscript. However, there are still serious doughts about the results. In one hand, the presented IC50 values do not correlate with the inhibition results obtained in the original assays, for expample at 500 uM the best inhibition (about 85 %) was obtained by IBunDM while both BTBnDM and BFBnDM gave about 77 % inhibition for 3CLpro, the corresponding IC50 values from Table S3 are 243, 427 and 99 uM, repcectively. Based on these only about 50 % inhibition is expected at 500 uM of BTBnDM, much higher degree thatn 77 % for BFBnDM. Even more worrying the now corrected CC50 values, originally given in uM now stated in mM. There is no meaining of 107 mM CC50 value (Table S5) calculated from measurements up to 1 mM (Figure S8).